# Swimming eukaryotic microorganisms exhibit a universal speed distribution

**Maciej Lisicki[1,2†]\*, Marcos F Velho Rodrigues[1†], Raymond E Goldstein[1], Eric Lauga[1]\***

[1]Department of Applied Mathematics and Theoretical Physics, University of Cambridge, Cambridge, United Kingdom; [2]Institute of Theoretical Physics, Faculty of Physics, University of Warsaw, Warsaw, Poland

**Abstract** One approach to quantifying biological diversity consists of characterizing the statistical distribution of specific properties of a taxonomic group or habitat. Microorganisms living in fluid environments, and for whom motility is key, exploit propulsion resulting from a rich variety of shapes, forms, and swimming strategies. Here, we explore the variability of swimming speed for unicellular eukaryotes based on published data. The data naturally partitions into that from flagellates (with a small number of flagella) and from ciliates (with tens or more). Despite the morphological and size differences between these groups, each of the two probability distributions of swimming speed are accurately represented by log-normal distributions, with good agreement holding even to fourth moments. Scaling of the distributions by a characteristic speed for each data set leads to a collapse onto an apparently universal distribution. These results suggest a universal way for ecological niches to be populated by abundant microorganisms.
DOI: https://doi.org/10.7554/eLife.44907.001

**\*For correspondence:**
maciej.lisicki@fuw.edu.pl (ML);
e.lauga@damtp.cam.ac.uk (EL)

[†]These authors contributed equally to this work

## Introduction

Unicellular eukaryotes comprise a vast, diverse group of organisms that covers virtually all environments and habitats, displaying a menagerie of shapes and forms. Hundreds of species of the ciliate genus *Paramecium* (*Wichterman, 1986*) or flagellated *Euglena* (*Buetow, 2011*) are found in marine, brackish, and freshwater reservoirs; the green algae *Chlamydomonas* is distributed in soil and fresh water world-wide (*Harris et al., 2009*); parasites from the genus *Giardia* colonize intestines of several vertebrates (*Adam, 2001*). One of the shared features of these organisms is their motility, crucial for nutrient acquisition and avoidance of danger (*Bray, 2001*). In the process of evolution, single-celled organisms have developed in a variety of directions, and thus their rich morphology results in a large spectrum of swimming modes (*Cappuccinelli, 1980*).

Many swimming eukaryotes actuate tail-like appendages called flagella or cilia in order to generate the required thrust (*Sleigh, 1975*). This is achieved by actively generating deformations along the flagellum, giving rise to a complex waveform. The flagellar axoneme itself is a bundle of nine pairs of microtubule doublets surrounding two central microtubules, termed the '9 + 2' structure (*Nicastro et al., 2005*), and cross-linking dynein motors, powered by ATP hydrolysis, perform mechanical work by promoting the relative sliding of filaments, resulting in bending deformations.

Although eukaryotic flagella exhibit a diversity of forms and functions (*Moran et al., 2014*), two large families, 'flagellates' and 'ciliates', can be distinguished by the shape and beating pattern of their flagella. Flagellates typically have a small number of long flagella distributed along the bodies, and they actuate them to generate thrust. The set of observed movement sequences includes planar undulatory waves and traveling helical waves, either from the base to the tip, or in the opposite direction (*Jahn and Votta, 1972*; *Brennen and Winet, 1977*). Flagella attached to the same body might follow different beating patterns, leading to a complex locomotion strategy that often relies

also on the resistance the cell body poses to the fluid. In contrast, propulsion of ciliates derives from the motion of a layer of densely-packed and collectively-moving cilia, which are short hair-like flagella covering their bodies. The seminal review paper of *Brennen and Winet (1977)* lists a few examples from both groups, highlighting their shape, beat form, geometric characteristics and swimming properties. Cilia may also be used for transport of the surrounding fluid, and their cooperativity can lead to directed flow generation. In higher organisms this can be crucial for internal transport processes, as in cytoplasmic streaming within plant cells (*Allen and Allen, 1978*), or the transport of ova from the ovary to the uterus in female mammals (*Lyons et al., 2006*).

Here, we turn our attention to these two morphologically different groups of swimmers to explore the variability of their propulsion dynamics within broad taxonomic groups. To this end, we have collected swimming speed data from literature for flagellated eukaryotes and ciliates and analyze them separately (we do not include spermatozoa since they lack (ironically) the capability to reproduce and are thus not living organisms; their swimming characteristics have been studied by *Tam and Hosoi, 2011*). A careful examination of the statistical properties of the speed distributions for flagellates and ciliates shows that they are not only both captured by log-normal distributions but that, upon rescaling the data by a characteristic swimming speed for each data set, the speed distributions in both types of organisms are essentially identical.

## Results and discussion

We have collected swimming data on 189 unicellular eukaryotic microorganisms ($N_{fl} = 112$ flagellates and $N_{cil} = 77$ ciliates) (see Appendix 1 and *Source data 1*). *Figure 1* shows a tree encompassing the phyla of organisms studied and sketches of a representative organism from each phylum. A large morphological variation is clearly visible. In addition, we delineate the branches involving aquatic organisms and parasitic species living within hosts. Both groups include ciliates and flagellates.

Due to the morphological and size differences between ciliates and flagellates, we investigate separately the statistical properties of each. *Figure 2* shows the two swimming speed histograms superimposed, based on the raw distributions shown in *Figure 2—figure supplement 1*, where bin widths have been adjusted to their respective samples using the Freedman-Diaconis rule (see Materials and methods). Ciliates span a much larger range of speeds, up to 7 mm/s, whereas generally smaller flagellates remain in the sub-mm/s range. The inset shows that the number of flagella in both groups leads to a clear division. To compare the two groups further, we have also collected information on the characteristic sizes of swimmers from the available literature, which we list in Appendix 1. The average cell size differs fourfold between the populations (31 µm for flagellates and 132 µm for ciliates) and the distributions, plotted in *Figure 2—figure supplement 2*, are biased towards the low-size end but they are quantitatively different. In order to explore the physical conditions, we used the data on sizes and speeds to compute the Reynolds number $\mathrm{Re} = UL/\nu$ for each organism, where $\nu = \eta/\rho$ is the kinematic viscosity of water, with $\eta$ the viscosity and $\rho$ the density. Since almost no data was available for the viscosity of the fluid in swimming speed measurements, we assumed the standard value $\nu = 10^{-6} m^2/s$ for water for all organisms. The distribution of Reynolds numbers (*Figure 2—figure supplement 3*), shows that ciliates and flagellates operate in different ranges of $\mathrm{Re}$, although for both groups $\mathrm{Re}<1$, imposing on them the same limitations of inertialess Stokes flow (*Purcell, 1977*; *Lauga and Powers, 2009*).

Furthermore, studies of green algae (*Short et al., 2006*; *Goldstein, 2015*) show that an important distinction between the smaller, flagellated species and the largest multicellular ones involves the relative importance of advection and diffusion, as captured by the Péclet number $Pe = UL/D$, where $L$ is a typical organism size and $D$ is the diffusion constant of a relevant molecular species. Using the average size $L$ of the cell body in each group of the present study ($L_{fl} = 31 \ \mu m$, $L_{cil} = 132\mu m$) and the median swimming speeds ($U_{fl} = 127\mathrm{m/s}$, $U_{cil} = 784\mathrm{m/s}$), and taking $D = 10^3 (\mu\mathrm{m})^2/\mathrm{s}$, we find $Pe_{fl} \sim 3.9$ and $Pe_{cil} \sim 103$, which further justifies analyzing the groups separately; they live in different physical regimes.

Examination of the mean, variance, kurtosis, and higher moments of the data sets suggest that the probabilities $P(U)$ of the swimming speed are well-described by log-normal distributions,

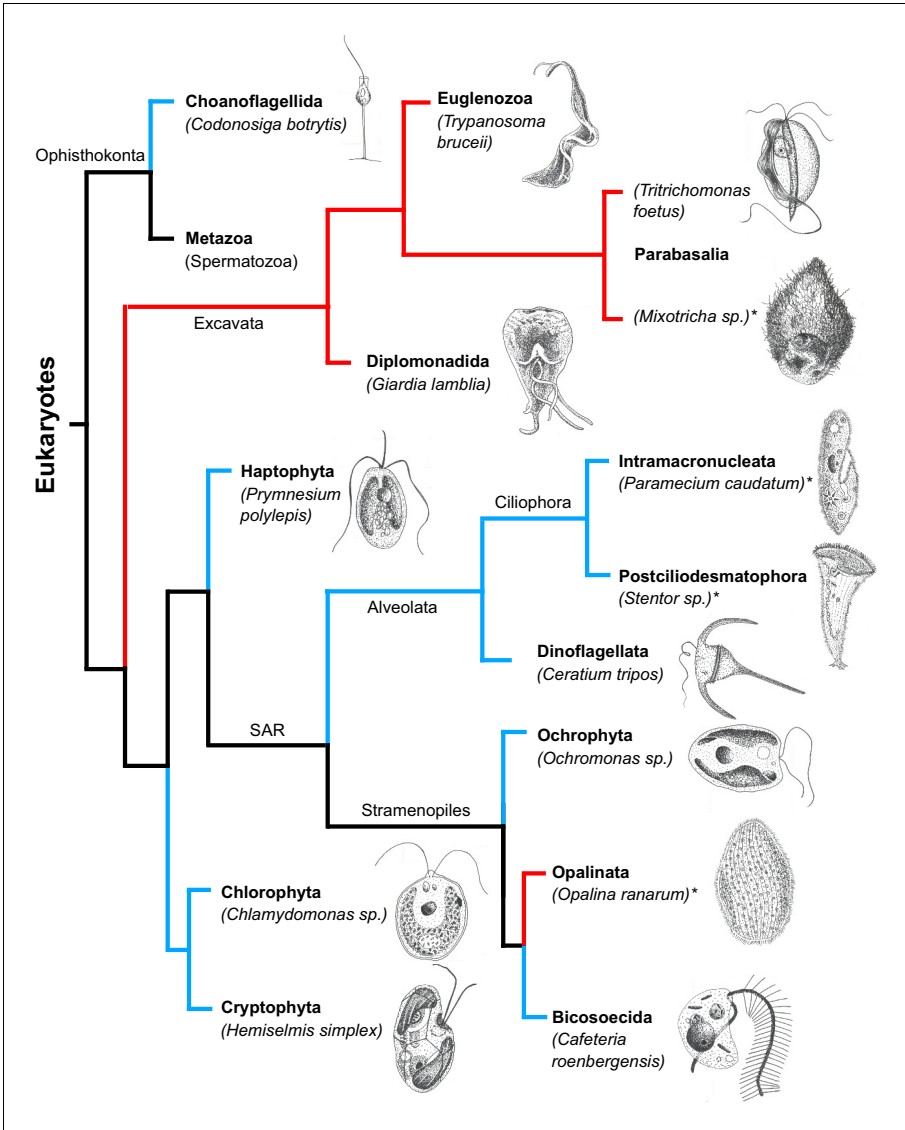

**Figure 1.** The tree of life (cladogram) for unicellular eukaryotes encompassing the phyla of organisms analyzed in the present study. Aquatic organisms (living in marine, brackish, or freshwater environments) have their branches drawn in blue while parasitic organisms have their branches drawn in red. Ciliates are indicated by an asterisk after their names. For each phylum marked in bold font, a representative organism has been sketched next to its name. Phylogenetic data from *Hinchliff et al. (2015)*.

DOI: https://doi.org/10.7554/eLife.44907.002

$$P(U) = \frac{1}{U\sigma\sqrt{2\pi}}\exp\left(-\frac{(\ln U - \mu)^2}{2\sigma^2}\right), \tag{1}$$

normalized as $\int_0^\infty dU P(U) = 1$, where $\mu$ and $\sigma$ are the mean and the standard deviation of $\ln U$. The median $M$ of the distribution is $e^\mu$, with units of speed. Log-normal distributions are widely observed across nature in areas such as ecology, physiology, geology and climate science, serving as an empirical model for complex processes shaping a system with many potentially interacting elements (*Limpert et al., 2001*), particularly when the underlying processes involve proportionate fluctuations or multiplicative noise (*Koch, 1966*).

The results of fitting (see Materials and methods) are plotted in *Figure 3*, where the best fits are presented as solid curves, with the shaded areas representing 95% confidence intervals. For

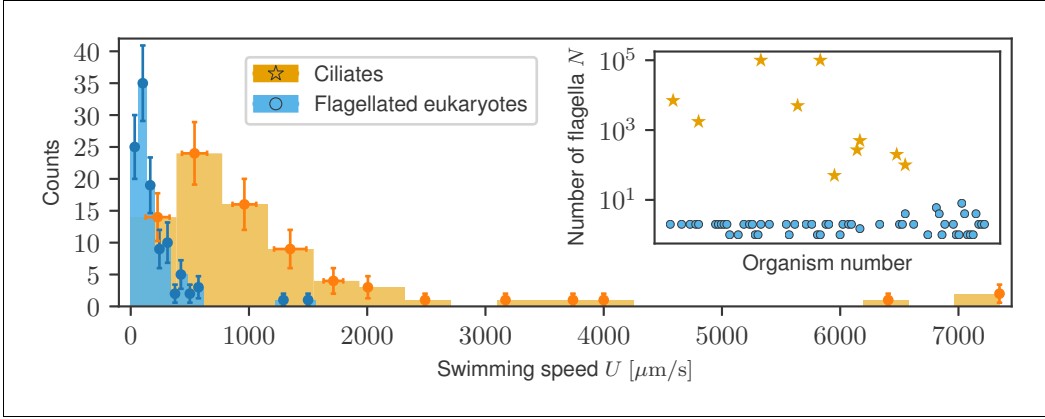

**Figure 2.** Histograms of swimming speed for ciliates and flagellates demonstrate a similar character but different scales of velocities. Data points represent the mean and standard deviation of the data in each bin; horizontal error bars represent variability within each bin, vertical error bars show the standard deviation of the count. Inset: number of flagella displayed, where available, for each organism exhibits a clear morphological division between ciliates and flagellates.

DOI: https://doi.org/10.7554/eLife.44907.003

The following figure supplements are available for figure 2:

**Figure supplement 1.** Linear distribution of swimming speed data.

DOI: https://doi.org/10.7554/eLife.44907.004

**Figure supplement 2.** Distribution of organism sizes in analyzed groups.

DOI: https://doi.org/10.7554/eLife.44907.005

**Figure supplement 3.** Distribution of Reynolds numbers for organisms in analyzed groups.

DOI: https://doi.org/10.7554/eLife.44907.006

flagellates, we find the $M_{fl} = 127 \text{m/s}$ and $\sigma_{fl} = 0.978$ while for ciliates, we obtain $M_{cil} = 784 \text{m/s}$ and $\sigma_{cil} = 0.936$. Log-normal distributions are known to emerge from an (imperfect) analogy to the Gaussian central limit theorem (see Materials and methods). Since the data are accurately described by this distribution, we conclude that the published literature includes a sufficiently large amount of unbiased data to be able to see the whole distribution.

We next compare the statistical variability within groups by examining rescaled distributions (*Goldstein, 2018*). As each has a characteristic speed $M$, we align the peaks by plotting the distributions versus the variable $U/M$ for each group. Since $P$ has units of 1/speed, we are thus led to the form $P(U, M) = M^{-1}F(U/M)$ for some function $F$. For the log-normal distribution, with $M$ the median, we find

$$F(\xi) = \frac{1}{\xi\sigma\sqrt{2\pi}}\exp\left(-\frac{\ln^2\xi}{2\sigma^2}\right), \tag{2}$$

which now depends on the single parameter $\sigma$ and has a median of unity by construction. To study the similarity of the two distributions we plot the functions $F = MP(U/M)$ for each. As seen in *Figure 4*, the rescaled distributions are essentially indistinguishable, and this can be traced back to the near identical values of the variances $\sigma$, which are within 5% of each other. The fitting uncertainties shown shaded in *Figure 4* suggest a very similar range of variability of the fitted distributions. Furthermore, both the integrated absolute difference between the distributions (0.028) and the Kullback-Leibler divergence (0.0016) are very small (see Materials and methods), demonstrating the close similarity of the two distributions. This similarity is robust to the choice of characteristic speed, as shown in *Figure 4—figure supplement 1*, where the arithmetic mean $U^*$ is used in place of the median.

In living cells, the sources for intrinsic variability within organisms are well characterized on the molecular and cellular level (*Kirkwood et al., 2005*) but less is known about variability within taxonomic groups. By dividing unicellular eukaryotes into two major groups on the basis of their difference in morphology, size and swimming strategy, we were able to capture in this paper the log-

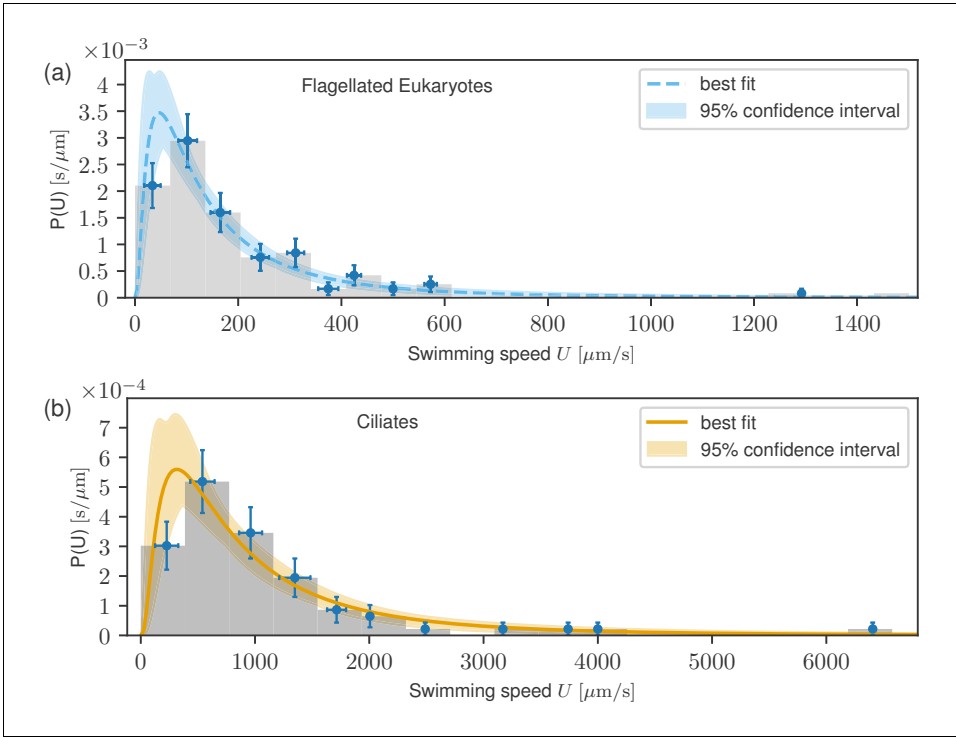

**Figure 3.** Probability distribution functions of swimming speeds for flagellates (**a**) and ciliates (**b**) with the fitted log-normal distributions. Data points represent uncertainties as in *Figure 2*. Despite the markedly different scales of the distributions, they have similar shapes.

DOI: https://doi.org/10.7554/eLife.44907.007

The following figure supplement is available for figure 3:

**Figure supplement 1.** Higher moments of the swimming speed distributions obtained from the data compared with those calculated from the fitted log-normal distribution.

DOI: https://doi.org/10.7554/eLife.44907.008

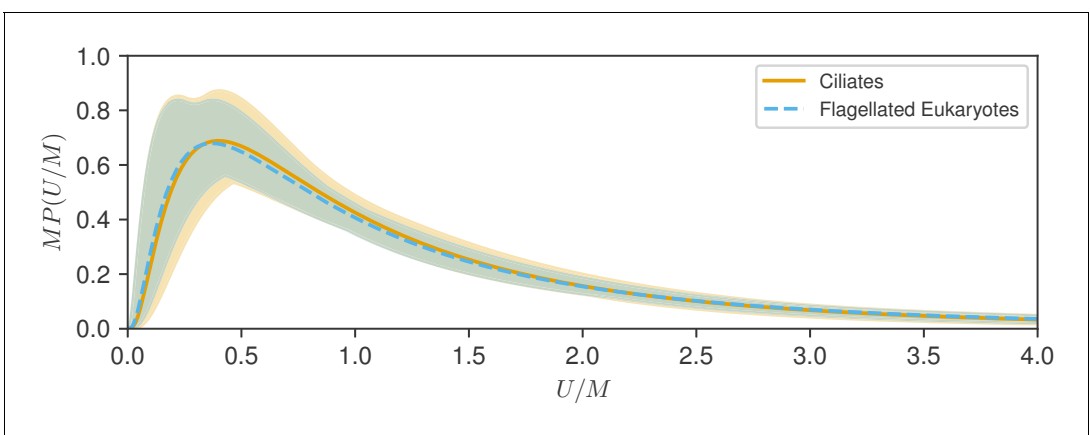

**Figure 4.** Test of rescaling hypothesis. Shown are the two fitted log-normal curves for flagellates and ciliates, each multiplied by the distribution median $M$, plotted versus speed normalized by $M$. The distributions for show remarkable similarity and uncertainty of estimation.

DOI: https://doi.org/10.7554/eLife.44907.009

The following figure supplement is available for figure 4:

**Figure supplement 1.** Data collapse as in the main figure, but using the mean speeds $U^*$ instead of the median $M$.

DOI: https://doi.org/10.7554/eLife.44907.010

normal variability within each subset. Using a statistical analysis of the distributions as functions of the median swimming speed for each population we further found an almost identical distribution of swimming speeds for both types of organisms. Our results suggest that the observed log-normal randomness captures a universal way for ecological niches to be populated by abundant microorganisms with similar propulsion characteristics. We note, however, that the distributions of swimming speeds among species do not necessarily reflect the distributions of swimming speeds among individuals, for which we have no available data.

## Materials and methods

### Data collection

Data for ciliates were sourced from 26 research articles, while that for flagellates were extracted from 48 papers (see Appendix 1). Notably, swimming speeds reported in the various studies have been measured under different physiological and environmental conditions, including temperature, viscosity, salinity, oxygenation, pH and light. Therefore we consider the data *not* as representative of a uniform environment, but instead as arising from a random sampling of a wide range of environmental conditions. In cases where no explicit figure was given for $U$ in a paper, estimates were made using other available data where possible. Size of swimmers has also been included as a characteristic length for each organism. This, however, does not reflect the spread and diversity of sizes within populations of individual but is rather an indication of a typical size, as in the considered studies these data were not available. Information on anisotropy (different width/length) is also not included.

No explicit criteria were imposed for the inclusion in the analyses, apart from the biological classification (i.e. whether the organisms were unicellular eukaryotic ciliates/flagellates). We have used all the data found in literature for these organisms over the course of an extensive search. Since no selection was made, we believe that the observed statistical properties are representative for these groups.

### Data processing and fitting the log-normal distribution

Bin widths in histograms in *Figure 2* and *Figure 3* have been chosen separately for ciliates and flagellated eukaryotes according to the Freedman-Diaconis rule (*Freedman and Diaconis, 1981*) taking into account the respective sample sizes and the spread of distributions. The bin width $b$ is then given by the number of observations $N$ and the interquartile range of the data $\mathrm{IQR}$ as

$$b = 2\frac{\mathrm{IQR}}{N^{1/3}}. \tag{3}$$

Within each bin in *Figure 3*, we calculate the mean and the standard deviation for the binned data, which constitute the horizontal error bars. The vertical error bars reflect the uncertainty in the number of counts $N_j$ in bin $j$. This is estimated to be Poissonian, and thus the absolute error amounts to $\sqrt{N_j}$. Notably, the relative error decays with the number of counts as $1/\sqrt{N_j}$.

In fitting the data, we employ the log-normal distribution *Equation (1)*. In general, from from data comprising $N$ measurements, labelled $x_i$ ($i = 1, ..., N$), the $n$-th arithmetic moment $\mathcal{M}_n$ is the expectation $\mathbb{E}(X^n)$, or

$$\mathcal{M}_n = \frac{1}{N}\sum_{i=1}^{N} x_i^n \tag{4}$$

Medians of the data were found by sorting the list of values and picking the middlemost value. For a log-normal distribution, the arithmetic moments are given solely by $\mu$ and $\sigma$ of the associated normal distribution as

$$\mathcal{M}_n = M^n \Sigma^{n^2}, \tag{5}$$

where we have defined $M = \exp(\mu)$ and $\Sigma = \exp(\sigma^2/2)$, and note that $M$ is the median of the

distribution. Thus, the mean is $M\Sigma$ and the variance is $M^2\Sigma^2(\Sigma^2-1)$. From the first and second moments, we estimate

$$\mu = \ln\left(\frac{\mathcal{M}_1^2}{\sqrt{\mathcal{M}_2}}\right) \quad \text{and} \quad \sigma^2 = \ln\left(\frac{\mathcal{M}_2}{\mathcal{M}_1^2}\right). \tag{6}$$

Having estimated $\mu$ and $\sigma$, we can compute the higher order moments from *Equation (5)* and compare to those calculated directly from the data, as shown in *Figure 3—figure supplement 1*.

To fit the data, we have used both the MATLAB fitting routines and the Python scipy.stats module. From these fits we estimated the shape and scale parameters and the 95% confidence intervals in *Figure 3* and *Figure 4*. We emphasize that the fitting procedures use the raw data via the maximum likelihood estimation method, and not the processed histograms, hence the estimated parameters are insensitive to the binning procedure.

For rescaled distributions, the average velocity for each group of organisms was calculated as $U^* = \frac{1}{N_i}\sum_{i=1}^{N_i} U_i$, with $i \in \{cil, fl\}$. Then, data in each subset have been rescaled by the area under the fitted curve to ensure that the resulting probability density functions $p_i$ are normalized as

$$\int_0^\infty p_i(x)\mathrm{d}x = 1. \tag{7}$$

In characterizations of biological or ecological diversity, it is often assumed that the examined variables are Gaussian, and thus the distribution of many uncorrelated variables attains the normal distribution by virtue of the Central Limit Theorem (CLT). In the case when random variables in question are positive and have a log-normal distribution, no analogous explicit analytic result is available. Despite that, there is general agreement that a sum of independent log-normal random variables can be well approximated by another log-normal random variable. It has been proven by *Szyszkowicz and Yanikome (2009)* that the sum of identically distributed equally and positively correlated joint log-normal distributions converges to a log-normal distribution of known characteristics but for uncorrelated variables only estimations are available (*Beaulieu et al., 1995*). We use these results to conclude that our distributions contain enough data to be unbiased and seen in full.

## Comparisons of distributions

In order to quantify the differences between the fitted distributions, we define the integrated absolute difference $\Delta$ between two probability distributions $p(x)$ and $q(x)$ ($x > 0$) as

$$\Delta = \int_0^\infty |p(x) - q(x)|\mathrm{d}x. \tag{8}$$

As the probability distributions are normalized, this is a measure of their relative 'distance'. As a second measure, we use the Kullback-Leibler divergence (*Kullback and Leibler, 1951*),

$$D(p,q) = \int_0^\infty p(x)\ln\left(\frac{p(x)}{q(x)}\right)\mathrm{d}x. \tag{9}$$

Note that $D(p,q) \neq D(q,p)$ and therefore $D$ is not a distance metric in the space of probability distributions.

## Acknowledgements

This project has received funding from the European Research Council (ERC) under the European Union's Horizon 2020 research and innovation program (grant agreement 682754 to EL), and from Established Career Fellowship EP/M017982/1 from the Engineering and Physical Sciences Research Council and Grant 7523 from the Gordon and Betty Moore Foundation (REG).

## Additional information

### Competing interests
Raymond E Goldstein: Reviewing editor, *eLife*. The other authors declare that no competing interests exist.

### Funding

| Funder | Grant reference number | Author |
| --- | --- | --- |
| H2020 European Research Council | 682754 | Eric Lauga |
| Engineering and Physical Sciences Research Council | EP/M017982/ | Raymond E Goldstein |
| Gordon and Betty Moore Foundation | 7523 | Raymond E Goldstein |

The funders had no role in study design, data collection and interpretation, or the decision to submit the work for publication.

### Author contributions
Maciej Lisicki, Conceptualization, Data curation, Software, Formal analysis, Validation, Investigation, Methodology, Writing—original draft, Writing—review and editing; Marcos F Velho Rodrigues, Data curation, Software, Formal analysis, Investigation, Visualization, Methodology, Writing—original draft, Writing—review and editing; Raymond E Goldstein, Investigation, Methodology, Writing—review and editing; Eric Lauga, Conceptualization, Formal analysis, Supervision, Funding acquisition, Validation, Investigation, Methodology, Writing—original draft, Project administration, Writing—review and editing

### Author ORCIDs
Maciej Lisicki https://orcid.org/0000-0002-6976-0281
Marcos F Velho Rodrigues https://orcid.org/0000-0002-8744-6966
Raymond E Goldstein https://orcid.org/0000-0003-2645-0598
Eric Lauga https://orcid.org/0000-0002-8916-2545

### Decision letter and Author response
Decision letter https://doi.org/10.7554/eLife.44907.016
Author response https://doi.org/10.7554/eLife.44907.017

## Additional files

### Supplementary files
• Source data 1. Spreadsheet data for swimming eukaryotes listed in Appendix 1 and Appendix 2.
DOI: https://doi.org/10.7554/eLife.44907.011

• Transparent reporting form
DOI: https://doi.org/10.7554/eLife.44907.012

### Data availability
All data generated or analysed during this study are included in the manuscript.

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

# Appendix 1

DOI: https://doi.org/10.7554/eLife.44907.013

The Appendix contains the data which form the basis of our study. The tables contain data on the sizes and swimming speed of ciliates organisms and flagellated eukaryotes from the existing literature. Data for ciliates were sourced from 26 research articles, while data for the flagellates were extracted from 48 papers. In the cases where two or more sources reported contrasting figures for the swimming speed, the average value is reported in our tables. The data itself is available in *Source data 1*.

## Data for swimming flagellates

Abbreviations: dflg. – dinoflagellata; dph – dinophyceae; chlph. – chlorophyta; ochph. (het.) – ochrophyta (heterokont); srcm. – sarcomastigophora, pyr. – pyramimonadophyceae; prym. – prymnesiophyceae; dict. – dictyochophyceae; crypt. – cryptophyceae; chrys. – chrysophyceae

| Species | Phylum | Class | $L[\mu m]$ | $U[\mu m/s]$ | References |
|---|---|---|---|---|---|
| *Alexandrium minutum* | dflg. | dph. | 21.7 | 222.5 | (*Lewis et al., 2006*) |
| *Alexandrium ostenfeldii* | dflg. | dph. | 41.1 | 110.5 | (*Lewis et al., 2006*) |
| *Alexandrium tamarense* | dflg. | dph. | 26.7 | 200 | (*Lewis et al., 2006*) |
| *Amphidinium britannicum* | dflg. | dph. | 51.2 | 68.7 | (*Bauerfeind et al., 1986*) |
| *Amphidinium carterae* | dflg. | dph. | 16 | 81.55 | (*Gittleson et al., 1974*; *Bauerfeind et al., 1986*) |
| *Amphidinium klebsi* | dflg. | dph. | 35 | 73.9 | (*Gittleson et al., 1974*) |
| *Apedinella spinifera* | ochph. (het.) | dict. | 8.25 | 132.5 | (*Throndsen, 1973*) |
| *Bodo designis* | euglenozoa | kinetoplastea | 5.5 | 39 | (*Visser and Kiørboe, 2006*) |
| *Brachiomonas submarina* | chlph. | chlorophyceae | 27.5 | 96 | (*Bauerfeind et al., 1986*) |
| *Cachonina (Heterocapsa) niei* | dflg. | dph. | 21.4 | 302.8 | (*Levandowsky and Kaneta, 1987*; *Kamykowski and Zentara, 1977*) |
| *Cafeteria roenbergensis* | bygira (heterokont) | bicosoecida | 2 | 94.9 | (*Fenchel and Blackburn, 1999*) |
| *Ceratium cornutum* | dflg. | dph. | 122.3 | 177.75 | (*Levandowsky and Kaneta, 1987*; *Metzner, 1929*) |
| *Ceratium furca* | dflg. | dph. | 122.5 | 194 | (*Peters, 1929*) |
| *Ceratium fusus* | dflg. | dph. | 307.5 | 156.25 | (*Peters, 1929*) |
| *Ceratium hirundinella* | dflg. | dph. | 397.5 | 236.1 | (*Levandowsky and Kaneta, 1987*) |
| *Ceratium horridum* | dflg. | dph. | 225 | 20.8 | (*Peters, 1929*) |
| *Ceratium lineatus* | dflg. | dph. | 82.1 | 36 | (*Fenchel, 2001*) |
| *Ceratium longipes* | dflg. | dph. | 210 | 166 | (*Peters, 1929*) |
| *Ceratium macroceros* | dflg. | dph. | 50 | 15.4 | (*Peters, 1929*) |
| *Ceratium tripos* | dflg. | dph. | 152.3 | 121.7 | (*Peters, 1929*; *Bauerfeind et al., 1986*) |

*continued*

| Species | Phylum | Class | L[μm] | U[μm/s] | References |
|---------|--------|-------|-------|---------|------------|
| Chilomonas paramecium | cryptophyta | crypt. | 30 | 111.25 | (*Lee, 1954*; *Jahn and Bovee, 1967*; *Gittleson et al., 1974*) |
| Chlamydomonas reinhardtii | chlph. | chlorophyceae | 10 | 130 | (*Gittleson et al., 1974*; *Roberts, 1981*; *Guasto et al., 2010*) |
| Chlamydomonas moewusii | chlph. | chlorophyceae | 12.5 | 128 | (*Gittleson et al., 1974*) |
| Chlamydomonas sp. | chlph. | chlorophyceae | 13 | 63.2 | (*Lowndes, 1944*; *Lowndes, 1941*; *Bauerfeind et al., 1986*) |
| Crithidia deanei | euglenozoa | kinetoplastea | 7.4 | 45.6 | (*Gadelha et al., 2007*) |
| Crithidia fasciculata | euglenozoa | kinetoplastea | 11.1 | 54.3 | (*Gadelha et al., 2007*) |
| Crithidia (Strigomonas) oncopelti | euglenozoa | kinetoplastea | 8 .1 | 18.5 | (*Roberts, 1981*; *Gittleson et al., 1974*) |
| Crypthecodinium cohnii | dflg. | dph. | n/a | 122.8 | (*Fenchel, 2001*) |
| Dinophysis acuta | dflg. | dph. | 65 | 500 | (*Peters, 1929*) |
| Dinophysis ovum | dflg. | dph. | 45 | 160 | (*Buskey et al., 1993*) |
| Dunaliella sp. | chlph. | chlorophyceae | 10.8 | 173.5 | (*Gittleson et al., 1974*; *Bauerfeind et al., 1986*) |
| Euglena gracilis | euglenozoa | euglenida (eugl.) | 47.5 | 111.25 | (*Lee, 1954*; *Jahn and Bovee, 1967*; *Gittleson et al., 1974*) |
| Euglena viridis | euglenozoa | euglenida (eugl.) | 58 | 80 | (*Holwill, 1975*; *Roberts, 1981*; *Lowndes, 1941*) |
| Eutreptiella gymnastica | euglenozoa | euglenida (aphagea) | 23.5 | 237.5 | (*Throndsen, 1973*) |
| Eutreptiella sp. R | euglenozoa | euglenida | 50 | 135 | (*Throndsen, 1973*) |
| Exuviaella baltica (Prorocentrum balticum) | dflg. | dph. | 15.5 | 138.9 | (*Wheeler, 1966*) |
| Giardia lamblia | srcm. | zoomastigophora | 11.25 | 26 | (*Lenaghan et al., 2011*; *Campanati et al., 2002*; *Chen et al., 2012*) |
| Gonyaulax polyedra | dflg. | dph. | 39.2 | 254.05 | (*Hand et al., 1965*; *Gittleson et al., 1974*; *Kamykowski et al., 1992*) |
| Gonyaulax polygramma | dflg. | dph. | 46.2 | 500 | (*Levandowsky and Kaneta, 1987*) |
| Gymnodinium aureolum | dflg. | dph. | n/a | 394 | (*Meunier et al., 2013*) |
| Gymnodinium sanguineum (splendens) | dflg. | dph. | 47.6 | 220.5 | (*Kamykowski et al., 1992*; *Levandowsky and Kaneta, 1987*) |
| Gymnodinium simplex | dflg. | dph. | 10.6 | 559 | (*Jakobsen et al., 2006*) |
| Gyrodinium aureolum | dflg. | dph. | 30.5 | 139 | (*Bauerfeind et al., 1986*; *Throndsen, 1973*) |

*continued*

| Species | Phylum | Class | $L$[$\mu$m] | $U$[$\mu$m/s] | References |
|---|---|---|---|---|---|
| *Gyrodinium dorsum* (bi-flagellated) | dflg. | dph. | 37.5 | 324 | (*Hand et al., 1965*; *Gittleson et al., 1974*; *Kamykowski et al., 1992*; *Levandowsky and Kaneta, 1987*; *Brennen and Winet, 1977*) |
| *Gyrodinium dorsum* (uni-flagellated) | dflg. | dph. | 34.5 | 148.35 | (*Hand and Schmidt, 1975*) |
| *Hemidinium nasutum* | dflg. | dph. | 27.2 | 105.6 | (*Levandowsky and Kaneta, 1987*; *Metzner, 1929*) |
| *Hemiselmis simplex* | cryptophyta | crypt. | 5.25 | 325 | (*Throndsen, 1973*) |
| *Heterocapsa pygmea* | dflg. | dph. | 13.5 | 102.35 | (*Bauerfeind et al., 1986*) |
| *Heterocapsa rotundata* | dflg. | dph. | 12.5 | 323 | (*Jakobsen et al., 2006*) |
| *Heterocapsa triquetra* | dflg. | dph. | 17 | 97 | (*Visser and Kiørboe, 2006*) |
| *Heteromastix pyriformis* | chlph. | nephrophyseae | 6 | 87.5 | (*Throndsen, 1973*) |
| *Hymenomonas carterae* | haptophyta | prym. | 12.5 | 87 | (*Bauerfeind et al., 1986*) |
| *Katodinium rotundatum (Heterocapsa rotundata)* | dflg. | dph. | 10.8 | 425 | (*Levandowsky and Kaneta, 1987*; *Throndsen, 1973*) |
| *Leishmania major* | euglenozoa | kinetoplastea | 12.5 | 36.4 | (*Gadelha et al., 2007*) |
| *Menoidium cultellus* | euglenozoa | euglenida (eugl.) | 45 | 136.75 | (*Holwill, 1975*; *Votta et al., 1971*) |
| *Menoidium incurvum* | euglenozoa | euglenida (eugl.) | 25 | 50 | (*Lowndes, 1941*; *Gittleson et al., 1974*) |
| *Micromonas pusilla* | chlph. | mamiellophyceae | 2 | 58.5 | (*Bauerfeind et al., 1986*; *Throndsen, 1973*) |
| *Monas stigmata* | ochph. (het.) | chrys. | 6 | 269 | (*Gittleson et al., 1974*) |
| *Monostroma angicava* | chlph. | ulvophyceae | 6.7 | 170.55 | (*Togashi et al., 1997*) |
| *Nephroselmis pyriformis* | chlph. | nephrophyseae | 4.8 | 163.5 | (*Bauerfeind et al., 1986*) |
| *Oblea rotunda* | dflg. | dph. | 20 | 420 | (*Buskey et al., 1993*) |
| *Ochromonas danica* | ochph. (het.) | chrys. | 8.7 | 77 | (*Holwill and Peters, 1974*) |
| *Ochromonas malhamensis* | ochph. (het.) | chrys. | 3 | 57.5 | (*Holwill, 1974*) |
| *Ochromonas minima* | ochph. (het.) | chrys. | 5 | 75 | (*Throndsen, 1973*) |
| *Olisthodiscus luteus* | ochph. (het.) | raphidophyceae | 22.5 | 90 | (*Bauerfeind et al., 1986*; *Throndsen, 1973*) |
| *Oxyrrhis marina* | dflg. | oxyrrhea | 39.5 | 300 | (*Boakes et al., 2011*; *Fenchel, 2001*) |
| *Paragymnodinium shiwhaense* | dflg. | dph. | 10.9 | 571 | (*Meunier et al., 2013*) |
| *Paraphysomonas vestita* | ochph. (het.) | chrys. | 14.7 | 116.85 | (*Christensen-Dalsgaard and Fenchel, 2004*) |

*continued*

| Species | Phylum | Class | $L$[μm] | $U$[μm/s] | References |
|---|---|---|---|---|---|
| *Pavlova lutheri* | haptophyta | pavlovophyceae | 6.5 | 126 | (*Bauerfeind et al., 1986*) |
| *Peranema trichophorum* | euglenozoa | euglenida (heteronematales) | 45 | 20 | (*Lowndes, 1941*; *Gittleson et al., 1974*; *Brennen and Winet, 1977*) |
| *Peridinium bipes* | dflg. | dph. | 42.9 | 291 | (*Fenchel, 2001*) |
| *Peridinium cf. quinquecorne* | dflg. | dph. | 19 | 1500 | (*Bauerfeind et al., 1986*; *Levandowsky and Kaneta, 1987*; *Horstmann, 1980*) |
| *Peridinium cinctum* | dflg. | dph. | 47.5 | 120 | (*Bauerfeind et al., 1986*; *Levandowsky and Kaneta, 1987*; *Metzner, 1929*) |
| *Peridinium (Protoperidinium) claudicans* | dflg. | dph. | 77.5 | 229 | (*Peters, 1929*) |
| *Peridinium (Protoperidinium) crassipes* | dflg. | dph. | 102 | 100 | (*Peters, 1929*) |
| *Peridinium foliaceum* | dflg. | dph. | 30.6 | 185.2 | (*Kamykowski et al., 1992*) |
| *Peridinium (Bysmatrum) gregarium* | dflg. | dph. | 32.5 | 1291.7 | (*Levandowsky and Kaneta, 1987*) |
| *Peridinium (Protoperidinium) ovatum* | dflg. | dph. | 61 | 187.5 | (*Peters, 1929*) |
| *Peridinium (Peridiniopsis) penardii* | dflg. | dph. | 28.8 | 417 | (*Sibley et al., 1974*) |
| *Peridinium (Protoperidinium) pentagonum* | dflg. | dph. | 92.5 | 266.5 | (*Peters, 1929*) |
| *Peridinium (Protoperidinium) subinerme* | dflg. | dph. | 50 | 285 | (*Peters, 1929*) |
| *Peridinium trochoideum* | dflg. | dph. | 25 | 53 | (*Levandowsky and Kaneta, 1987*) |
| *Peridinium umbonatum* | dflg. | dph. | 30 | 250 | (*Levandowsky and Kaneta, 1987*; *Metzner, 1929*) |
| *Phaeocystis pouchetii* | haptophyta | prym. | 6.3 | 88 | (*Bauerfeind et al., 1986*) |
| *Polytoma uvella* | chlph. | chlorophyceae | 22.5 | 100.9 | (*Lowndes, 1944*; *Gittleson et al., 1974*; *Lowndes, 1941*) |
| *Polytomella agilis* | chlph. | chlorophyceae | 12.4 | 150 | (*Gittleson and Jahn, 1968*; *Gittleson and Noble, 1973*; *Gittleson et al., 1974*; *Roberts, 1981*) |
| *Prorocentrum mariae-lebouriae* | dflg. | dph. | 14.8 | 141.05 | (*Kamykowski et al., 1992*; *Bauerfeind et al., 1986*; *Miyasaka et al., 1998*) |
| *Prorocentrum micans* | dflg. | dph. | 45 | 329.1 | (*Bauerfeind et al., 1986*; *Levandowsky and Kaneta, 1987*) |
| *Prorocentrum minimum* | dflg. | dph. | 15.1 | 107.7 | (*Bauerfeind et al., 1986*; *Miyasaka et al., 1998*) |
| *Prorocentrum redfieldii Bursa (P. triestinum)* | dflg. | dph. | 33.2 | 333.3 | (*Sournia, 1982*) |

*continued*

| Species | Phylum | Class | L[μm] | U[μm/s] | References |
|---|---|---|---|---|---|
| *Protoperidinium depressum* | dflg. | dph. | 132 | 450 | (*Buskey et al., 1993*) |
| *Protoperidinium granii (Ostf.) Balech* | dflg. | dph. | 57.5 | 86.1 | (*Sournia, 1982*) |
| *Protoperidinium pacificum* | dflg. | dph. | 54 | 410 | (*Buskey et al., 1993*) |
| *Prymnesium polylepis* | haptophyta | prym. | 9.1 | 45 | (*Dölger et al., 2017*) |
| *Prymnesium parvum* | haptophyta | prym. | 7.2 | 30 | (*Dölger et al., 2017*) |
| *Pseudopedinella pyriformis* | ochph. (het.) | dict. | 6.5 | 100 | (*Throndsen, 1973*) |
| *Pseudoscourfieldia marina* | chlph. | pyr. | 4.1 | 42 | (*Bauerfeind et al., 1986*) |
| *Pteridomonas danica* | ochph. (het.) | dict. | 5.5 | 179.45 | (*Christensen-Dalsgaard and Fenchel, 2004*) |
| *Pyramimonas amylifera* | chlph. | pyr. | 24.5 | 22.5 | (*Bauerfeind et al., 1986*) |
| *Pyramimonas cf. disomata* | chlph. | pyr. | 9 | 355 | (*Throndsen, 1973*) |
| *Rhabdomonas spiralis* | euglenozoa | euglenida (aphagea) | 27 | 120 | (*Holwill, 1975*) |
| *Rhodomonas salina* | cryptophyta | crypt. | 14.5 | 588.5 | (*Jakobsen et al., 2006*; *Meunier et al., 2013*) |
| *Scrippsiella trochoidea* | dflg. | dph. | 25.3 | 87.6 | (*Kamykowski et al., 1992*; *Bauerfeind et al., 1986*; *Sournia, 1982*) |
| *Spumella* sp. | ochph. (het.) | chrys. | 10 | 25 | (*Visser and Kiørboe, 2006*) |
| *Teleaulax* sp. | cryptophyta | crypt. | 13.5 | 98 | (*Meunier et al., 2013*) |
| *Trypanosoma brucei* | euglenozoa | kinetoplastea | 18.8 | 20.5 | (*Rodríguez et al., 2009*) |
| *Trypanosoma cruzi* | euglenozoa | kinetoplastea | 20 | 172 | (*Jahn and Fonseca, 1963*; *Brennen and Winet, 1977*) |
| *Trypanosoma vivax* | euglenozoa | kinetoplastea | 23.5 | 29.5 | (*Bargul et al., 2016*) |
| *Trypanosoma evansi* | euglenozoa | kinetoplastea | 21.5 | 16.1 | (*Bargul et al., 2016*) |
| *Trypanosoma congolense* | euglenozoa | kinetoplastea | 18 | 9.7 | (*Bargul et al., 2016*) |
| *Tetraflagellochloris mauritanica* | chlph. | chlorophyceae | 4 | 300 | (*Barsanti et al., 2016*) |

# Appendix 2

DOI: https://doi.org/10.7554/eLife.44907.013

## Data for swimming ciliates

Abbreviations: imnc. = intramacronucleata; pcdph. = postciliodesmatophora; olig. – oligohymenophorea; spir. – spirotrichea; hettr. – heterotrichea; lit. – litostomatea; eugl. – euglenophyceae

| Species | Phylum | Class | $L[\mu m]$ | $U[\mu m/s]$ | References |
|---|---|---|---|---|---|
| *Amphileptus gigas* | imnc. | lit. | 808 | 608 | (*Bullington, 1925*) |
| *Amphorides quadrili-neata* | imnc. | spir. | 138 | 490 | (*Buskey et al., 1993*) |
| *Balanion comatum* | imnc. | prostomatea | 16 | 220 | (*Visser and Kiørboe, 2006*) |
| *Blepharisma* | pcdph. | hettr. | 350 | 600 | (*Sleigh and Blake, 1977*; *Roberts, 1981*) |
| *Coleps hirtus* | imnc. | prostomatea | 94.5 | 686 | (*Bullington, 1925*) |
| *Coleps* sp. | imnc. | prostomatea | 78 | 523 | (*Bullington, 1925*) |
| *Colpidium striatum* | imnc. | olig. | 77 | 570 | (*Beveridge et al., 2010*) |
| *Condylostoma patens* | pcdph. | hettr. | 371 | 1061 | (*Bullington, 1925*; *Machemer, 1974*) |
| *Didinium nasutum* | imnc. | lit. | 140 | 1732 | (*Bullington, 1925*; *Machemer, 1974*; *Roberts, 1981*; *Sleigh and Blake, 1977*) |
| *Euplotes charon* | imnc. | spir. | 66 | 1053 | (*Bullington, 1925*) |
| *Euplotes patella* | imnc. | spir. | 202 | 1250 | (*Bullington, 1925*) |
| *Euplotes vannus* | imnc. | spir. | 82 | 446 | (*Wang et al., 2008*; *Ricci et al., 1997*) |
| *Eutintinnus cf. pinguis* | imnc. | spir. | 147 | 410 | (*Buskey et al., 1993*) |
| *Fabrea salina* | pcdph. | hettr. | 184.1 | 216 | (*Marangoni et al., 1995*) |
| *Favella panamensis* | imnc. | spir. | 238 | 600 | (*Buskey et al., 1993*) |
| *Favella* sp. | imnc. | spir. | 150 | 1080 | (*Buskey et al., 1993*) |
| *Frontonia* sp. | imnc. | olig. | 378.5 | 1632 | (*Bullington, 1925*) |
| *Halteria grandinella* | imnc. | spir. | 50 | 533 | (*Bullington, 1925*; *Gilbert, 1994*) |
| *Kerona polyporum* | imnc. | spir. | 107 | 476.5 | (*Bullington, 1925*) |
| *Laboea strobila* | imnc. | spir. | 100 | 810 | (*Buskey et al., 1993*) |
| *Lacrymaria lagenula* | imnc. | lit. | 45 | 909 | (*Bullington, 1925*) |
| *Lembadion bullinum* | imnc. | olig. | 43 | 415 | (*Bullington, 1925*) |
| *Lembus velifer* | imnc. | olig. | 87 | 200 | (*Bullington, 1925*) |
| *Mesodinium rubrum* | imnc. | lit. | 38 | 7350 | (*Jonsson and Tiselius, 1990*; *Riisgård and Larsen, 2009*; *Crawford and Lindholm, 1997*) |
| *Metopides contorta* | imnc. | armophorea | 115 | 359 | (*Bullington, 1925*) |
| *Nassula ambigua* | imnc. | nassophorea | 143 | 2004 | (*Bullington, 1925*) |
| *Nassula ornata* | imnc. | nassophorea | 282 | 750 | (*Bullington, 1925*) |

continued

| Species | Phylum | Class | $L$[μm] | $U$[μm/$s$] | References |
|---|---|---|---|---|---|
| Opalina ranarum | placidozoa (heterokont) | opalinea | 350 | 50 | (**Blake, 1975**; **Sleigh and Blake, 1977**) |
| Ophryoglena sp. | imnc. | olig. | 325 | 4000 | (**Machemer, 1974**) |
| Opisthonecta henneg | imnc. | olig. | 126 | 1197 | (**Machemer, 1974**; **Jahn and Hendrix, 1969**) |
| Oxytricha bifara | imnc. | spir. | 282 | 1210 | (**Bullington, 1925**) |
| Oxytricha ferruginea | imnc. | spir. | 150 | 400 | (**Bullington, 1925**) |
| Oxytricha platystoma | imnc. | spir. | 130 | 520 | (**Bullington, 1925**) |
| Paramecium aurelia | imnc. | olig. | 244 | 1650 | (**Bullington, 1925**; **Bullington, 1930**) |
| Paramecium bursaria | imnc. | olig. | 130 | 1541.5 | (**Bullington, 1925**; **Bullington, 1930**) |
| Paramecium calkinsii | imnc. | olig. | 124 | 1392 | (**Bullington, 1930**; **Bullington, 1925**) |
| Paramecium cauda-tum | imnc. | olig. | 225.5 | 2489.35 | (**Bullington, 1930**; **Jung et al., 2014**) |
| Paramecium marinum | imnc. | olig. | 115 | 930 | (**Bullington, 1925**) |
| Paramecium multimi-cronucleatum | imnc. | olig. | 251 | 3169.5 | (**Bullington, 1930**) |
| Paramecium polycaryum | imnc. | olig. | 91 | 1500 | (**Bullington, 1930**) |
| Paramecium spp. | imnc. | olig. | 200 | 975 | (**Jahn and Bovee, 1967**; **Sleigh and Blake, 1977**; **Roberts, 1981**) |
| Paramecium tetraurelia | imnc. | olig. | 124 | 784 | (**Funfak et al., 2015**) |
| Paramecium woodruf-fi | imnc. | olig. | 160 | 2013.5 | (**Bullington, 1930**) |
| Porpostoma notatum | imnc. | olig. | 107.7 | 1842.2 | (**Fenchel and Blackburn, 1999**) |
| Prorodon teres | imnc. | prostomatea | 175 | 1066 | (**Bullington, 1925**) |
| Spathidium spathula | imnc. | lit. | 204.5 | 526 | (**Bullington, 1925**) |
| Spirostomum ambiguum | pcdph. | hettr. | 1045 | 810 | (**Bullington, 1925**) |
| Spirostomum sp. | pcdph. | hettr. | 1000 | 1000 | (**Sleigh and Blake, 1977**) |
| Spirostomum teres | pcdph. | hettr. | 450 | 640 | (**Bullington, 1925**) |
| Stenosemella steinii | imnc. | spir. | 83 | 190 | (**Buskey et al., 1993**) |
| Stentor caeruleus | pcdph. | hettr. | 528.5 | 1500 | (**Bullington, 1925**) |
| Stentor polymorphus | pcdph. | hettr. | 208 | 887 | (**Bullington, 1925**; **Sleigh and Aiello, 1972**; **Sleigh, 1968**) |
| Strobilidium spiralis | imnc. | spir. | 60 | 330 | (**Buskey et al., 1993**) |
| Strobilidium velox | imnc. | spir. | 43 | 150 | (**Gilbert, 1994**) |
| Strombidinopsis acuminatum | imnc. | spir. | 80 | 390 | (**Buskey et al., 1993**) |
| Strombidium clapare-di | imnc. | spir. | 69.5 | 3740 | (**Bullington, 1925**) |
| Strombidium conicum | imnc. | spir. | 75 | 570 | (**Buskey et al., 1993**) |
| Strombidium sp. | imnc. | spir. | 33 | 360 | (**Buskey et al., 1993**) |

*continued*

| Species | Phylum | Class | $L$[$\mu$m] | $U$[$\mu$m/$s$] | References |
|---|---|---|---|---|---|
| *Strombidium sulcatum* | imnc. | spir. | 32.5 | 995 | (*Fenchel and Jonsson, 1988*; *Fenchel and Blackburn, 1999 Fenchel and Blackburn, 1999*) |
| *Stylonichia sp.* | imnc. | spir. | 167 | 737.5 | (*Bullington, 1925*; *Machemer, 1974*) |
| *Tetrahymena pyriformis* | imnc. | olig. | 72.8 | 475.6 | (*Sleigh and Blake, 1977*; *Roberts, 1981*; *Brennen and Winet, 1977*) |
| *Tetrahymena thermophila* | imnc. | olig. | 46.7 | 204.5 | (*Wood et al., 2007*) |
| *Tillina magna* | imnc. | colpodea | 162.5 | 2000 | (*Bullington, 1925*) |
| *Tintinnopsis kofoidi* | imnc. | spir. | 100 | 400 | (*Buskey et al., 1993*) |
| *Tintinnopsis minuta* | imnc. | spir. | 40 | 60 | (*Buskey et al., 1993*) |
| *Tintinnopsis tubulosa* | imnc. | spir. | 95 | 160 | (*Buskey et al., 1993*) |
| *Tintinnopsis vasculum* | imnc. | spir. | 82 | 250 | (*Buskey et al., 1993*) |
| *Trachelocerca olor* | pcdph. | karyorelictea | 267.5 | 900 | (*Bullington, 1925*) |
| *Trachelocerca tenuicollis* | pcdph. | karyorelictea | 432 | 1111 | (*Bullington, 1925*) |
| *Uroleptus piscis* | imnc. | spir. | 203 | 487 | (*Bullington, 1925*) |
| *Uroleptus rattulus* | imnc. | spir. | 400 | 385 | (*Bullington, 1925*) |
| *Urocentrum turbo* | imnc. | olig. | 90 | 700 | (*Bullington, 1925*) |
| *Uronema filificum* | imnc. | olig. | 25.7 | 1372.7 | (*Fenchel and Blackburn, 1999*) |
| *Uronema marinum* | imnc. | olig. | 56.9 | 1010 | (*Fenchel and Blackburn, 1999*) |
| *Uronema sp.* | imnc. | olig. | 25 | 1175 | (*Sleigh and Blake, 1977*; *Roberts, 1981*) |
| *Uronychia transfuga* | imnc. | spir. | 118 | 6406 | (*Leonildi et al., 1998*) |
| *Uronychia setigera* | imnc. | spir. | 64 | 7347 | (*Leonildi et al., 1998*) |
| *Uronemella spp.* | imnc. | olig. | 28 | 250 | (*Petroff et al., 2015*) |

