## [Decision Letter]

Thank you for submitting your article "Swimming eukaryotic microorganisms exhibit a universal speed distribution" for consideration by *eLife*. Your article has been reviewed by two peer reviewers, and the evaluation has been overseen by Arup Chakraborty as the Senior and Reviewing Editor. The following individual involved in review of your submission has agreed to reveal his identity: Matthew Herron (Reviewer #1).

The reviewers have discussed the reviews with one another and the Reviewing Editor has drafted this decision to help you prepare a revised submission.

Summary:

This study analyzes published data on swimming speed among flagellated and ciliated microorganisms. Data from both groups fit similar, but separate, log normal distributions, leading the authors to infer a "universal way for ecological niches to be populated by abundant microorganisms." This short report is well-written, well-sourced, and has sufficient detail provided by the authors to replicate the findings. This methodology has the potential to enrich parallel genetics-based studies and provide deeper insights into the connections between ecology and evolution. However, we feel that the main claims of the paper need to be toned down or further substantiated in a revised submission.

Essential revisions:

1) The ecological implication of the results, expressed in the Abstract and at the end of the Discussion, concern how ecological niches are partitioned by microorganisms. The limitation of the analyzed data is that it is per species, and all species are treated equally, with no information about abundance. It is undoubtedly an important result that aquatic ecosystems contain many species that swim slowly and far fewer that swim quickly, at least within each category. However, without information about abundances, this result says very little about the distribution of swimming speeds within a given ecosystem; that is, the data are equally compatible with slow swimmers being rare (but diverse) and fast swimmers being common (but homogenous) or the opposite. So, the distribution of swimming speeds among all individuals is unclear. Similarly, flagellates could be common and ciliates rare or vice versa. The point that the distributions of swimming speeds among species do not necessarily reflect the distributions of swimming speeds among individuals needs to be addressed.

2) Can universality really be claimed with a set of just two distributions? Figures 2, 3, and 4 well-illustrate the authors' point, but comprise fitting and rescaling of just these two sets of data. The demonstration in Figure 4 that two log-normal distributions may be collapsed onto one another is not too surprising. It seems with such a data set on hand, there is much potential to explore a variety of questions that can enrich or explain the authors conclusions. In particular, addressing a few of the following points will enhance the paper:

• Did the authors examine any other distributions, such as cell size?

• Did the authors attempt a physical rescaling of the distributions, e.g. in terms of the Reynolds or Peclet numbers?

• Is there sufficient resolution in the data set (i.e. number of samples) to explore/compare subsets of the data such as uniflagellates versus multiflagellates and ciliates?

• Similar to the above comment, examining the swimming speed distributions of other taxonomic groups from the data set – i.e. corresponding to an early bifurcation in Figure 1 – may provide additional insights and strengthen the authors' argument for universality in the swimming speed distribution.

• While perhaps outside of the scope of the present short report, the potential implications of the authors' observation abound for other organismal systems. Did they consider examining other dataset, e.g. for higher organisms as in Gazzola et al., Nature Physics 10 (2014)?

---

## [Author Response]

Essential revisions:1] The ecological implication of the results, expressed in the Abstract and at the end of the Discussion, concern how ecological niches are partitioned by microorganisms. The limitation of the analyzed data is that it is per species, and all species are treated equally, with no information about abundance. It is undoubtedly an important result that aquatic ecosystems contain many species that swim slowly and far fewer that swim quickly, at least within each category. However, without information about abundances, this result says very little about the distribution of swimming speeds within a given ecosystem; that is, the data are equally compatible with slow swimmers being rare (but diverse) and fast swimmers being common (but homogenous) or the opposite. So, the distribution of swimming speeds among all individuals is unclear. Similarly, flagellates could be common and ciliates rare or vice versa. The point that the distributions of swimming speeds among species do not necessarily reflect the distributions of swimming speeds among individuals needs to be addressed.

This is a very interesting point and we agree with the reviewers. Alas, since there is little or no available data for abundance, it is difficult to make claim concerning particular ecosystems. We are implicitly assuming in our paper that the sampling of the underlying distributions was random. Of course, in real ecosystems there are interactions (symbiotic, mutualistic, etc.) among species so they are not necessarily “independent”. To state our point clearly, we have added a sentence in the last paragraph of the main text.

2] Can universality really be claimed with a set of just two distributions? Figures 2, 3, and 4 well-illustrate the authors' point, but comprise fitting and rescaling of just these two sets of data. The demonstration in Figure 4 that two log-normal distributions may be collapsed onto one another is not too surprising. It seems with such a data set on hand, there is much potential to explore a variety of questions that can enrich or explain the authors conclusions. In particular, addressing a few of the following points will enhance the paper:• Did the authors examine any other distributions, such as cell size?

Yes we have. We have now added to the paper the available data on cell sizes that we believe helps present a broader picture. Figure 2—figure supplement 2 contains histogram of cell sizes, produced using values given in the revised tables in Appendix 1. The cell sizes represent the 'characteristic' size for each cell (largest of the available sizes if different width/length were given in literature). The size distributions are distinct and no apparent similarity between them is visible. We have then used them to calculate the Reynolds and Péclet numbers, as suggested below.

• Did the authors attempt a physical rescaling of the distributions, e.g. in terms of the Reynolds or Peclet numbers?

Rescaled distributions have been added as Figure 2—figure supplement 3. We have plotted there the Reynolds number for each organism. Due to the paucity of reported viscosities in the analysed works, we assumed the viscosity to be that of water at standard conditions in each case. The distributions are different, since ciliates are generally larger and faster swimmers compared to flagellates. The Péclet number is proportional to the Reynolds number, since both contain the product of cell size and swimming velocity. Therefore, we choose one (Re) as a measure of the character of the fluid transport. We have modified the paper to indicate these points.

• Is there sufficient resolution in the data set (i.e. number of samples) to explore/compare subsets of the data such as uniflagellates versus multiflagellates and ciliates?

Unfortunately, the resolution of the subsets is not sufficient for the proposed comparison. The listed data was the information available in literature on the swimming problem. Our focus was to collect it and analyse it together. Moreover, some of the values given represent averages over samples analyzed in the papers, given with no further information on the uncertainty. In the plots, we have included the fitting errors and estimated the errors associated to the binning procedures. Within the available data, this statistical uncertainty was the only accessible measure.

• Similar to the above comment, examining the swimming speed distributions of other taxonomic groups from the data set – i.e. corresponding to an early bifurcation in Figure 1 – may provide additional insights and strengthen the authors' argument for universality in the swimming speed distribution.

Pointing back to the previous remark, we think the statistics for individual groups would not be sufficient to justify potential conclusions and thus we refrained from this analysis in the paper.

• While perhaps outside of the scope of the present short report, the potential implications of the authors' observation abound for other organismal systems. Did they consider examining other dataset, e.g. for higher organisms as in Gazzola et al., Nature Physics 10 (2014)?

The paper by Gazzola et al. concerns the relation that can be established between the Reynolds number for the flow created by a swimming organism and its swimming number, which involves the temporal details of the actuation (frequency of the periodic body motion which gives rise to the flow). In our study, we focus on unicellular microscale swimmers, and thus the Reynolds numbers rarely exceed 1 (see Figure 2—figure supplement 3), so the realm of higher Re is outside of the scope of the report.